# Netrin-1 in Glioblastoma Neovascularization: The New Partner in Crime?

**DOI:** 10.3390/ijms22158248

**Published:** 2021-07-31

**Authors:** Ximena Vásquez, Pilar Sánchez-Gómez, Verónica Palma

**Affiliations:** 1Laboratory of Stem Cells and Developmental Biology, Faculty of Sciences, Universidad de Chile, Santiago 7800003, Chile; ximevasquez283@gmail.com; 2Neurooncology Unit, Unidad Funcional de Investigación de Enfermedades Crónicas (UFIEC), Instituto de Salud Carlos III (ISCIII), 28220 Madrid, Spain

**Keywords:** glioblastoma, neovascularization, Netrin-1, exosomes, transdifferentiation, pericytes

## Abstract

Glioblastoma (GBM) is the most aggressive and common primary tumor of the central nervous system. It is characterized by having an infiltrating growth and by the presence of an excessive and aberrant vasculature. Some of the mechanisms that promote this neovascularization are angiogenesis and the transdifferentiation of tumor cells into endothelial cells or pericytes. In all these processes, the release of extracellular microvesicles by tumor cells plays an important role. Tumor cell-derived extracellular microvesicles contain pro-angiogenic molecules such as VEGF, which promote the formation of blood vessels and the recruitment of pericytes that reinforce these structures. The present study summarizes and discusses recent data from different investigations suggesting that Netrin-1, a highly versatile protein recently postulated as a non-canonical angiogenic ligand, could participate in the promotion of neovascularization processes in GBM. The relevance of determining the angiogenic signaling pathways associated with the interaction of Netrin-1 with its receptors is posed. Furthermore, we speculate that this molecule could form part of the microvesicles that favor abnormal tumor vasculature. Based on the studies presented, this review proposes Netrin-1 as a novel biomarker for GBM progression and vascularization.

## 1. Introduction

Glioblastoma (GBM) is the most aggressive tumor among gliomas, defined as primary brain cancers of a glial nature. Gliomas can be classified according to their histopathological resemblance as astrocytic tumors, oligodendrogliomas, and ependymomas. The World Health Organization classification specifies four grades of gliomas, of which grade IV corresponds to GBM. GBM is the most common of the primary astrocytomas and accounts for more than 60% of brain tumors in adults. Despite the use of radiotherapy and chemotherapy in patients who develop this type of tumor, they have a life expectancy that does not exceed 15 months and less than 5% of them survive 5 years after diagnosis [1].

GBM can emerge as a primary tumor, which develops de novo without previous clinical manifestations and affects mainly the elderly. They account for almost 90% of all GBM cases worldwide, and they have a worse prognosis than secondary tumors, which frequently affect younger patients. Primary GBM commonly exhibit amplification and/or mutations of the *Epidermal Growth Factor Receptor* (EGFR), mutations in the tumor suppressor genes *Phosphatase and Tensin Homolog* (*PTEN*) and *TP53*, complete loss of chromosome 10, deletion of *Cyclin Dependent Kinase Inhibitor 2A* (*CDKN2*), and mutations in the promoter of *Telomerase Reverse Transcriptase* (*TERT*). In secondary tumors, the most frequent alterations are mutations in *TP53* and the complete loss of chromosome 10 [2]. Mutations in *Isocitrate Dehydrogenase* (*IDH*) genes are more frequent in secondary GBM, where they reach 73% of the clinical cases, while in primary GBM, the percentage of *IDH* mutated tumors is barely 3.7% [3].

In GBM, there is infiltrative growth within healthy brain tissue and excessive angiogenesis that produces an abnormal tumor vasculature; both features are known as the main distinguishing characteristics of these tumors [4]. In the tumor microenvironment, communication between different cell groups and the promotion of some neovascularization processes in GBM occur thanks to the presence of molecules such as Vascular Endothelial Growth Factor (VEGF), basic Fibroblast Growth Factor (bFGF), Transforming Growth Factor beta (TGF-β), and Matrix Metalloproteinases (MMPs), either as soluble isoforms or contained in extracellular microvesicles released by tumor cells [5,6]. These signals result in the activation of different pathways involved in cell proliferation and migration, formation of new blood vessels, and transdifferentiation of tumor cells into endothelial cells (EC) and pericytes, culminating into a florid and robust tumor vasculature [7,8]. In addition to the participation of canonical angiogenic molecules, overexpression of Netrin-1, originally discovered as a neural guidance cue, has recently been described in GBM and an important role of this ligand in tumor angiogenesis has been proposed [9]. It has been suggested that upon interaction with its receptors, this ligand could trigger the activation of diverse signaling pathways that promote cell survival, proliferation, and migration. Moreover, Netrin-1 promotes the expression of C-MYC, which could favor neovascularization in the tumor niche [10].

The diversity of molecular alterations present in GBM has made it difficult to determine biomarkers and design effective pharmacological strategies to combat this tumor. One of the objectives of this review is to describe the mechanisms by which glioma cells interact with their cellular milieu to promote their aggressive growth. Moreover, we will suggest new markers that could be altered in GBM. In particular, we will highlight the role that Netrin-1 plays in the GBM microenvironment and suggest possible signaling pathways through which it could be participating in the promotion of neovascularization. A greater understanding of the molecular mechanisms underlying the onset and progression of GBM will allow for better diagnosis and development of more efficient therapeutic strategies for patients with this type of tumor.

## 2. Processes of Neovascularization in GBM

Different tumor neovascularization mechanisms have been described in GBM, such as vascular co-option, angiogenesis, vasculogenesis, vascular mimicry, and transdifferentiation of tumor cells into EC or pericytes [11]. In the process of vascular co-option, tumor cells move towards the blood vessels found in healthy tissue adjacent to the tumor, surround them, and move along the vessel wall [12]. This triggers events such as apoptosis, vascular regression, and necrosis, which lead to the initiation of angiogenesis: the formation of new blood vessels from existing ones [13]. Angiogenesis begins with the destabilization of the basement membrane of the blood vessels by breaking the contact between pericytes and EC. Next, the transition from an endothelial phenotype to a mesenchymal one occurs, hence promoting the proliferative, migratory, and invasive capacities of EC, which release different proteases that degrade the extracellular matrix and the basement membrane, allowing the proliferation of EC which give rise to the formation of new blood vessels [14].

Vasculogenesis occurs through the recruitment of bone marrow-derived endothelial precursor cells; also known as endothelial precursors, into the tumor tissue. Then, these cells differentiate and form de novo blood vessels [15].

During vascular mimicry, tumor cells carry out de novo formation of perfused blood vessels, which provide the tumor with sufficient blood and nutrients for its growth. Actually, GBM cells can transdifferentiate into EC and form functional blood vessels within the tumor, especially in its deepest areas [16]. It has been reported that part of the vascular endothelium in GBM has a neoplastic origin and that EC present genomic alterations similar to the tumor cells, hence originating from GBM stem cells (GSC) [17]. Furthermore, most of the pericytes generated in murine GBM models are derived as well from GSC and contribute to the formation of the tumor vasculature [18]. It has been proposed that the GSC acquire mural cells properties and contact with the EC, providing stability to the blood-brain-barrier (BBB). Cheng et al. demonstrated the ability of GSC to differentiate into pericytes in vitro and in vivo by verifying the expression of specific pericyte markers such as *alpha-Smooth Muscle Actin* (*α-SMA*), *Neuron-Glial antigen 2* (*NG2)*, *CD248*, and *CD146* [7]. They also showed that GSC-derived pericytes represent the highest percentage of these mural cells in GBM xenotransplants and that they are part of the new tumor vessels. These cells possessed the most common mutations present in tumor cells, such as *EGFR* amplification and *PTEN* loss, confirming that they came from GSC. Besides, this research confirmed that the elimination of tumor-derived pericytes disrupts the tumor vasculature, which suppresses GBM growth and progression. The authors described that GSC are recruited by tumor EC through Stromal Cell-Derived Factor 1 (SDF-1)/C-X-C Chemokine Receptor Type 4 (CXCR4) signaling and generate pericytes mainly through activation of the TGF-β pathway [7]. Others have shown that the process of transdifferentiation of GSC to pericytes, which allows the development of vascular mimicry, depends on the expression of the Vascular Endothelial Growth Factor Receptor 2 (VEGFR2/Flk-1) [18] or the activation of the Notch signaling pathway in the tumor cells [19]. Recent data from our group suggest that GSC-derived pericytes may be in contact with EC, acting as mural cells and enhancing the tumor vasculature, in GBM with *EGFR* mutations. Alternatively, they appear delocalized in overexpressing, but non-mutated *EGFR* tumors, favoring the disruption of the BBB [20,21]. These results suggest that these different behaviors could induce two distinct vascular phenotypes and explain the differential tumor aggressiveness of GBM with or without *EGFR* mutations. Moreover, our data indicates that the EGFR pathway controls the transdifferentiation of GSC from GBM to pericytes and that this process is mediated by the activation of Nuclear Factor kappa B (NF-κB) [20].

## 3. Signals Involved in GBM Angiogenesis

Some of the most studied canonical angiogenic molecules in GBM are proteins of the VEGF family, bFGF, Hepatocyte Growth Factor (HGF), Platelet-derived Growth Factor (PDGF), TGF-β, MMPs, and Angiopoietins [22] (Table 1). The VEGF family includes six secreted glycoproteins (VEGF-A, VEGF-B, VEGF-C, VEGF-D, VEGF-E, and placental growth factor PIGF), with VEGF-A being the best characterized. Increased expression of this gene is positively related to the tumor grade and the poor prognosis of glioma patients [23], both in lower-grade gliomas and GBM [21]. Notably, VEGF expression inversely correlates with the presence of *IDH* mutations, which identifies the less aggressive tumors. Moreover, among *IDH* wild-type GBMs, we have observed a higher expression of this and other angiogenic factors in *EGFR* mutant tumors, which also progress more rapidly than their wild-type counterparts [21].

There are different isoforms of VEGF-A, whose expression may be associated, for example, to the formation of permeable vasculature and the decrease in pericyte coverage [24]. VEGF signaling is mediated through binding to VEGFR-1 (Flt-1), VEGFR-2, and VEGFR-3, triggering angiogenesis, vascular permeability, and endothelial migration [25]. When VEGF binds to its receptors present on the EC membrane, they secrete MMPs, which degrade components of the extracellular matrix, thus promoting cell proliferation, migration, and invasion [26]. Besides, these ligands stimulate the proliferation and activation of pericytes, promoting their migration towards the new vessels and leading to their stabilization [27]. bFGF promotes angiogenesis by activating the proliferation and migration of EC and the indirect activation of cell migration. The population of GSC, whose stem cell state is maintained by bFGF, EGF, PDGF, or TGFβ [28,29,30,31], also promotes these processes. Meanwhile, HGF stimulates the proliferation and migration of EC and the overexpression of VEGF, in addition to suppressing the angiogenesis inhibitor, Thrombospondin 1 (TSP-1) [32]. PDGF directly induces EC proliferation and the formation of new blood vessels [33]. PDGF-B, a ligand that is part of the PDGF family of proteins, stimulates the proliferation of smooth muscle cells and pericytes towards the site where new vessels are forming and helps to establish a new basement membrane by binding to its PDGFR-β receptor [34]. Finally, angiopoietins are a family of four proteins (Ang-1-Ang-4) that fulfill different roles in angiogenesis. Ang-1 participates in the induction of new blood vessels formation and their stabilization through reciprocal interactions between the endothelium and the surrounding extracellular membrane [35]. Ang-2, which is up-regulated by hypoxia, increases the proliferation and migration of EC mediated by VEGF [36].

In addition to the molecules mentioned above, several non-classical angiogenic molecules, belonging to the Semaphorins, Ephrins, and Netrins families, mediate this process. All of these molecules were originally described as axon guidance molecules. However, their overexpression and participation in both pro and anti-angiogenic processes in cancer in general, and GBM in particular, has recently been demonstrated [37,38,39]. This review will describe in greater depth the role of Netrin-1 in the neovascularization process, the signaling pathways that could regulate its expression, and how this protein could be participating in the tumor microenvironment of GBM.

**Table 1 ijms-22-08248-t001:** Participation of classical angiogenic molecules in GBM.

Molecule	Function
VEGF	Promotes angiogenesis, vascular permeability, and endothelial migration [25].
MMPS	Degrade the components of the extracellular matrix, promote cell proliferation, migration and invasion [26]. Stimulate the proliferation and activation of pericytes and promote their migration towards the newly formed vessels [27].
BFGF	Promotes angiogenesis by activating EC proliferation and migration and activating cell migration. Maintains the stemness of GSC [28].
HGF	Stimulates the proliferation and migration of EC and the overexpression of VEGF. Suppresses the angiogenesis inhibitor, TSP-1 [32].
PDGF	Induces the proliferation of EC and the formation of new blood vessels [33]. Stimulates the proliferation of smooth muscle cells and pericytes towards the site where the new vessels are forming and helps to establish a new basement membrane by binding to its receptor PDGFR-β [34]. Promotes stem-like properties of GBM cells [31].
ANG-1	Participates in the induction of the formation of new blood vessels and their stabilization through reciprocal interactions between the endothelium and the surrounding extracellular membrane [35].
ANG-2	Increases VEGF-mediated proliferation and migration of EC [36].
SDF-1	Induces EC migration and proliferation [40].
HDGF	Promotes EC migration and angiogenesis in vivo [41].
FT	Induces an increase in the production of pro-angiogenic molecules such as HB-EGF through the activation of the PAR-2 receptor in EC, triggering an increase in the formation of tubules [42].

## 4. Expression of Netrin-1 and Its Receptors in GBM

Both in vitro and in vivo approaches have shown that Netrin-1 promotes tumor progression, metastasis, and angiogenesis in cancer [43]. Particularly, the expression of the Netrin-1 protein in GBM has been considered relevant in the promotion of tumor angiogenesis [44]. Notably, participation of the Netrin-1 signaling pathway in the angiogenesis processes in normal and injured brain has been reported [45,46].

Netrin-1 is part of the conserved family of Netrins, secreted and/or extracellular matrix proteins, whose original role in axonal guidance has been widely described [47]. The concentration of Netrin-1 in the brain in normal conditions varies between 50 and 150 ng/mL [48]. However, in pathological conditions such as GBM, the expression of this protein can increase between 2.1 to 4.5 times compared to normal tissues [10]. Different cell types in the tumor microenvironment in GBM such as glial cells (astrocytes and oligodendrocytes), microglia, infiltrating immune cells (monocytes, macrophages, and lymphocytes), EC, and pericytes can express Netrin-1 and its receptors [47,49,50,51,52,53,54,55,56,57,58] (Table 2). Immunohistochemical data from The Human Protein Atlas shows examples of positive Netrin-1 staining in glioma samples, with a diffuse pattern of expression (Figure 1A).

In axonal guidance, the chemo-attractant effects of Netrin-1 are mediated through Netrin-1 binding to the transmembrane molecules Deleted Colorectal Cancer (DCC) and Neogenin 1 (NEO1), while binding of Netrin-1 to receptors of the UNC5 (UNC5 A-D) family produces the repellents effects. Both DCC/NEO1 and UNC5 are so-called dependent receptors (DR). In the absence of their ligand, apoptosis is promoted. Their interaction with the ligand blocks apoptosis and stimulates cell survival, migration, and proliferation [59]. This is how DRs generate a state of cellular dependence in the presence of the specific ligand, in this case, Netrin-1. It has been shown that this molecule has a higher affinity for DCC/NEO1 receptor than for the UNC5 family of receptors [60] and that the effect produced by the binding of Netrin-1 to its receptors is different depending on the conformation they adopt [61]. The attractant effects on the axon are generated by interaction of the ligand with homodimers formed by either DCC and NEO1 receptors. The short-term repellent effects are generated by the binding of Netrin-1 to homodimers formed by UNC5 receptors, while the long-term effects are generated by the binding of the ligand to heterodimers formed by receptors of the DCC and/or NEO1 and UNC5 family [10,48,62].

Netrin-1 has been proposed as a possible biomarker for different types of cancer. Ramesh et al. demonstrated that Netrin-1 levels in the blood of patients with breast, kidney, prostate, liver, meningioma, pituitary adenoma cancer, and GBM are increased compared to control patients [63]. Notably, a comparative pan-cancer analysis shows a higher transcription of this gene in gliomas compared to other cancers (Figure 1B). Moreover, Netrin-1 expression is higher in type III and IV tumors, being related to cell aggressiveness and tumor progression [64]. In addition to the roles of Netrin-1 described above, the ligand has also been found to be a promoter of cellular invasiveness, which is mediated through the activation of Notch signaling [43,65]. Ylivinkka et al. determined that the activation of the Notch signaling pathway by Netrin-1 induces stemness properties on GBM cells [9]. Interestingly, the same authors described a stronger expression of Netrin-1 in areas surrounding necrosis in GBM samples, which are enriched in pro-angiogenic signals (9). In agreement with that, the in silico analysis shows a positive correlation between the expression of this ligand and different angiogenic-related genes such as *VEGFA*, *ANGPT-1*, and *ANGPT-2* (Figure 1C), supporting a possible role for Netrin-1 in the processes of glioma vascularization. Furthermore, it has been described that this molecule could participate in processes such as phagocytosis, the formation of adherent junctions, regulation of the actin cytoskeleton, and interactions between extracellular matrix and receptors [65].

**Figure 1 ijms-22-08248-f001:**
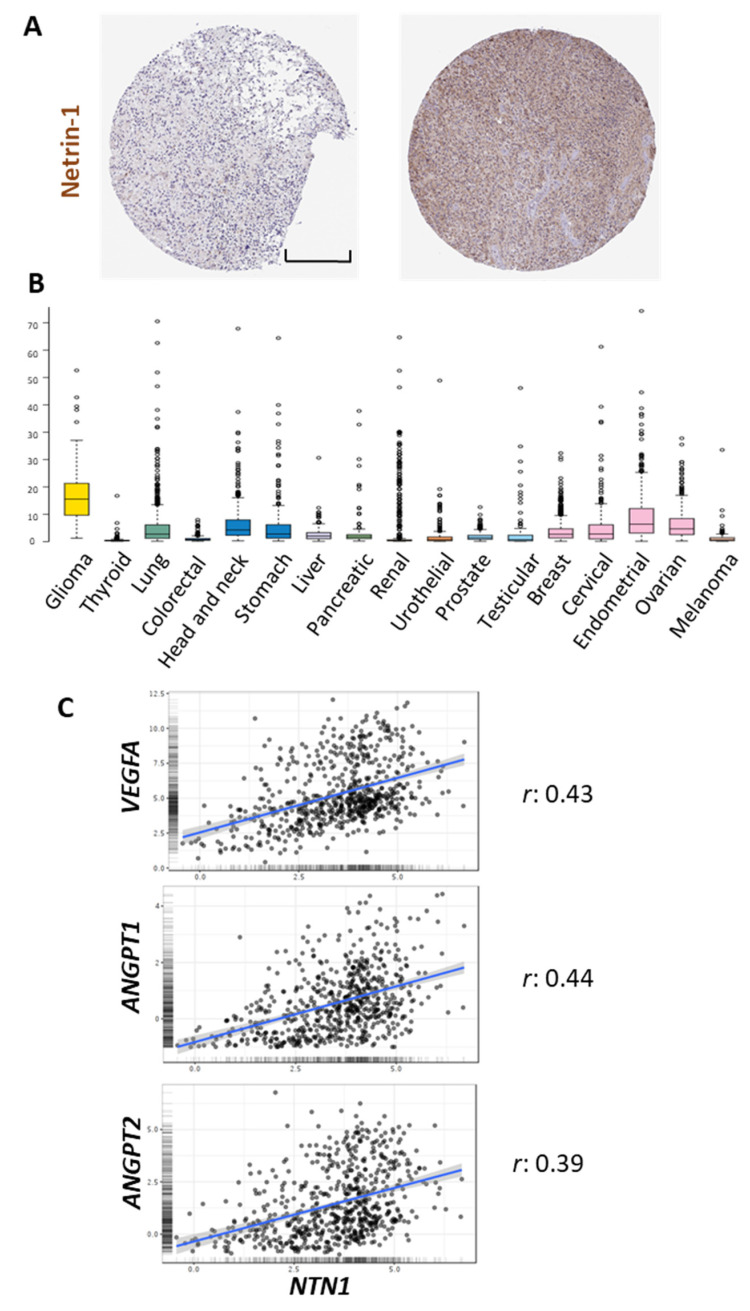
In silico analysis of NTN1 expression in gliomas. (**A**) Representative images of immunohistochemical staining of high-grade glioma samples using a Netrin-1 specific antibody. (**B**) RNA-seq data in 17 cancer types are reported as median FPKM (number Fragments Per Kilobase of exon per Million reads), generated by The Cancer Genome Atlas (TCGA). Data adapted under creative commons license from The Human Protein Atlas [66,67]. Images available from https://www.proteinatlas.org/ENSG00000065320-NTN1/pathology, accessed on 12 May 2021. (**C**) Pearson’s correlation analysis between the expression of *NTN1* and *VEGFA*, *ANGPT1*, *ANGPT2*, using data from the Chinese Glioma Genome Atlas (CGGA) cohort visualized in the GlioVis data portal [68] (http://gliovis.bioinfo.cnio.es, accessed on 12 May 2021). The correlation coefficient (*r*) is shown on the right. Scale bar in A: 200 μm.

Netrin-1 expression has been detected in different human GBM cell lines, such as the commonly used U87MG or the more aggressive and invasive U373MG and U251MG cell lines. Furthermore, it has been reported that the expression levels of Netrin-1 in human astrocytes are lower than in U373MG and U251MG, but higher than in U87 cells [10]. On the other hand, in brain endothelial cells (BEC), the expression of Netrin-1 has been described as moderate, while the expression of its receptor, NEO1, is high [58]. U251 cells express the UNC5A, UNC5B, UNC5C, and UNC5D receptors, while U87 and SHG44 cells express only the UNC5A and UNC5B receptors. NEO1, for its part, is expressed in GBM patient’s biopsies as well as in the GBM cell lines U87MG, U373MG, and U25 [64].

Netrin-1 isoform associated with a poor prognosis has been detected in neuroblastoma, bronchioalveolar carcinoma, breast and colon cancer [69]. The generation of this isoform occurs from an internal alternative promoter in the *NTN1* gene, which results in the formation of a truncated isoform of Netrin-1 that lacks the first part of the N-terminal VI domain and is located in the nucleolus. This protein interacts with ribosomal DNA promoter sequences, which stimulates ribosomal RNA synthesis, promoting cell proliferation and tumor growth [39,69]. Such a truncated form of Netrin-1 has not been described so far in GBM.

## 5. Signaling Pathways Associated with Netrin-1 Regulation

The participation of Netrin-1 in different biological processes has been widely described. However, it is still necessary to deepen the study of the mechanisms that regulate its expression. One of the signaling pathways involved in cancer and specifically in the development of GBM is activated by the Sonic Hedgehog (SHH) molecule [70,71]. The expression of components of this pathway such as SHH, PTCH1, GLI1, and GLI2 has been demonstrated in gliomas. Moreover, activation of this signaling cascade has been linked to proliferation, survival, self-renewal, and tumorigenicity in GBM cell lines [71]. In another brain tumor type, medulloblastoma, we have shown that *NEO1* is a direct target of the canonical SHH signaling pathway. Milla et al. described that in these tumors, the expression of *NEO-1* is transcriptionally upregulated by the SHH effector Gli2, through its binding to the promoter regions of the *NEO1* gene [72]. Meanwhile, the Gli1 effector seems to regulate directly the transcription of *NTN1* (Falcon et al., unpublished observations). The latter is particularly relevant since overexpression of GLI1 has been reported in GBM [72]. In addition, it has been recently described that Netrin-1 participates in the maintenance of the stability of the BBB, being a downstream effector of the SHH pathway in other types of CNS-related diseases [73].

Interestingly, activation of *NTN1* expression in GBM could also be promoted by NF-κB, in an indirect manner since it has been reported that this transcription factor induces the expression of SHH in cancer [74]. In addition, *NTN-1* promoter has been shown to be directly regulated by the NF-κB pathway in colorectal cancer, which allows us to suggest that a similar regulatory mechanism could occur in GBM (Figure 2) [75].

## 6. Signaling Pathways Associated with Netrin-1 and Angiogenesis

The interaction between Netrin-1 and its DRs produces the activation of different signaling pathways that promote processes such as cell migration, proliferation, and angiogenesis. It has recently been postulated that a downstream effector of this signaling could be *C-MYC* [10]. The expression of this gene is deregulated in GBM and its activation has been reported to promote increased expression of VEGF, inducing angiogenesis and tumor growth [76,77,78]. This regulation could occur through the regulation of the availability of VEGF and the angiogenesis inhibitor TSP-1 [79], the binding of C-MYC to the VEGF promoter region [78], or the regulation of IL-1β expression [80]. Whereas the connections of C-MYC with angiogenesis is well stablished, the activation of this transcription factor in response to Netrin-1 signaling remains to be fully characterized (Figure 2). Recent results obtained in our laboratory indicate that interaction between Netrin-1 and NEO1 triggers the activation of Integrin β1 (ITGB1) through the key integrin signaling component Focal Adhesion Kinase (FAK), which promotes cell migration and, consequently, metastasis in neuroblastoma [81]. Upon Netrin-1 binding to NEO1, FAK undergoes auto-phosphorylation at tyrosine residue 397 to induce intercellular signaling. Our findings are in line with previous work illustrating that DCC/NEO1 can coopt FAK to induce downstream signaling [82]. Of note, it has been described that the activation of ITGB1 through FAK and the regulation of MAPK/SRC pathways, in addition to promoting cell proliferation, induces the expression of C*-MYC* in tumor cells [83]. Moreover, a series of papers have indicated that netrins can bind directly to the α6β4 or α3ITGB1 integrin complex and cause integrin signaling. As such, evidence for cross talk—and potentially cooperative signaling—by integrins and classical Netrin-1 receptors in the context of GBM cannot be discarded. Hence, it is possible to suggest that the activation of the signaling complex Netrin-1/NEO-1 could lead to the activation of ITGB1 through FAK and subsequent activation of *C-MYC* in GBM, resulting in aberrant vascularization.

Interestingly, Huyge et al. described recently that the expression of Netrin-1 and its receptors NEO1 and UNC5B participate in the promotion of pluripotency in Mouse Embronic Stem Cells (mESCs) through co-regulation of the Wnt and MAPK signaling pathways. According to their results, increased expression of Netrin-1 causes a greater activation of the Wnt pathway through FAK, which promotes the phosphorylation of Gsk3α/β and the stabilization of β-catenin, processes in which the participation of receptors NEO1 and UNC5B is required. Moreover, the promotion of pluripotency in mESCs due to the expression of Netrin-1 is mediated in part by the induction of *C-MYC* activation and expression [84]. These results suggest that oncogenic activation of *C-MYC* could be mediated by the regulation of the Wnt/β-catenin signaling pathway, which is also activated in GBM, promoting proliferation processes, apoptosis inhibition, and cell invasion [85].

Finally, it has been described that the expression of *C-MYC* in GBM can also be induced through the interaction of Netrin-1 and the UNC5A receptor, and that this interaction could promote the growth of gliomas through the induction of *C-MYC*, mediated by activation of the NF-κB pathway [10]. As mentioned, NF-κB signaling is aberrantly activated in GBM and has been implicated in processes such as stem cell maintenance, stimulation of cell invasion, cell migration, angiogenesis, and resistance to radiation therapy [86]. Such signaling has been shown to regulate *C-MYC* expression in GBM through phosphorylation of Ser536 in the p65 subunit of NF-κB [10]. This subunit is part of the five molecules that make up the NF-κB family of transcription factors, which regulate the expression of a wide range of genes involved in proliferation, apoptosis, DNA repair, and immune response [87].

Based on the data presented, we suggest that Netrin-1 could participate in the control of neovascularization processes in GBM through the regulation of *C-MYC* expression by activating three possible signaling axes: NTN1/NEO1/FAK/Integrinβ1 axis, NTN1/UNC5B/NEO1/Gsk3α/β axis, and/or the NTN1/UNC5A/NF-κB axis (Figure 3). However, it would be necessary to determine the conformation that Netrin-1 receptors acquire in each of the proposed signaling pathways (homodimers or heterodimers of the respective receptors). Moreover, how tumor cells process, pack, and secrete Netrin-1 towards the tumor microenvironment has not been described either. This ligand could be released in a soluble way to the medium or, alternatively, contained within extracellular microvesicles.

## 7. Pro-Angiogenic Molecules Contained in Exosomes Derived from GBM Cells

The promotion of angiogenesis in the tumor microenvironment can also be carried out through the release of molecules contained in exosomes. Exosomes are microvesicles released by cells under both normal and pathological physiological conditions. They have a diameter of less than 100 nm, have a lipid bilayer and contain different molecules such as proteins, lipids, DNA, RNA, among others [88]. Different investigations reveal the presence of growth factors, proteases, miRNAs, and long non-coding RNAs in exosomes released by GBM cells [5,6,8,9,89,90,91]. All these molecules can promote angiogenesis in GBM through increased proliferative, migratory, invasive, and tubule-formation capacities of EC (Figure 4) [6,8,9,91,92,93]. Noteworthy, many of the angiogenic molecules released by GSC are found within exosomes and participate in the promotion of neovascularization processes [89,94,95,96].

As we have highlighted, tumor angiogenesis in GBM is also promoted by the expression of non-canonical angiogenic molecules such as Netrin-1. In GBM, overexpression of this protein has been identified both in cell lines and in tumor tissue and its participation in promoting tumor angiogenesis through increased cell proliferation and tubule formation has been reported. Classic angiogenic molecules such as VEGF-A have been identified in the conditioned medium of GBM cells [96]. However, there is still no information that reports the presence of Netrin-1 in conditioned medium or exosomes of GBM cells, and in particular, in GSC. Recently, the presence of axonal guidance molecules within exosomes released by tumor cells has been described [97], which would suggest that Netrin-1 could be released as well as contained within these microvesicles.

Although the presence of Netrin-1 in microvesicles derived from GBM tumor cells has not been reported to date, other molecules with similar functions have been found in exosomes. Saman et al. demonstrated that the overexpression of TAU in neuroblastoma cells promotes the enrichment of molecules associated with axonal guidance, such as netrins in exosomes released by those cells [98]. As Netrin-1 is also an axonal guidance protein and both the expression of TAU and Netrin-1 have been reported in GBM, it is possible to think that the expression of TAU could promote the enrichment of Netrin-1 in exosomes released by those cells. In GBM, the presence of proteins belonging to the Semaphorin family, axonal guidance proteins as well and non-canonical angiogenic proteins have been reported within exosomes [99]. Moreover, Semaphorin 7A has been reported to be exposed on the surface of exosomes and binds to GSC through ITGB1. This interaction activates FAK within GSC, promoting their mobility [100].

The angiogenic molecules released by GBM cells can also promote other neovascularization processes in GBM, such as the process of transdifferentiation of tumor cells to pericytes. Previously, in this review, the participation of GSC in the process of transdifferentiation of pericytes was highlighted, which through pro-angiogenic molecules such as VEGF and TGF-β promote the appearance of pericytes derived from GSC [101]. We could hypothesize that the secretion of exosomes containing Netrin-1 might participate in the promotion of the process of transdifferentiation from tumor cells to pericytes through its interaction with the UNC5A receptor, which would trigger the activation of NF-κB [69] and the tumor-to-pericyte transformation [20].

## 8. Projections

Data presented in this review strongly suggested a possible role of Netrin-1 in the neovascularization processes in GBM. As discussed earlier, the study of how GBM secretes Netrin-1 is primordial. In particular, it is required to determine if this protein is present in a soluble way in the conditioned medium of the tumor cells or within the microvesicles released by them. Furthermore, it would be necessary to determine if the Netrin-1 concentration in exosomes derived from GBM cells is higher than in exosomes released by other cell types present in the tumor microenvironment, such as astrocytes. The use of exosomes to find new cancer biomarkers is of great importance in the clinical area, because the methods used now, such as the detection of antigens in blood, are mostly for more than one type of cancer. In addition, techniques such as the blood test in stool samples in colorectal cancer and the endoscopic tests carried out for the diagnosis of gastric and colon cancer are highly invasive for patients, so the use of more specific and sensitive techniques, such as detection of markers contained in exosomes are necessary for cancer detection [102]. In GBM, currently, the diagnostic methods are based on neuroimaging techniques and brain biopsies, the latter being an invasive method for the patient. The detection of GBM biomarkers contained in exosomes from blood or urine samples would be a very useful non-invasive alternative [103].

In this way, a high concentration of Netrin-1 could be related to the presence of GBM and not to a normal condition. Furthermore, it would be interesting to determine if the angiogenic effect of Netrin-1 varies according to the way it is released. In addition to knowing the role of Netrin-1 in the angiogenesis process in GBM, it would be useful to study its participation in some of the neovascularization processes described above. By employing a co-culture of BEC cells with GSC, the ligands participation in the process of transdifferentiation of GSC to pericytes could be carried out. Specifically, the GSC’s Netrin-1 expression could be decreased through loss of function approaches, which would then be co-cultured with BEC cells. This would allow to determine if their ability to transdifferentiate into pericytes decreases and/or if their capacity to attach to ECs is affected. These studies could provide information on the participation of Netrin-1 in tumor angiogenesis processes, its role in the process of transdifferentiation of tumor cells to pericytes.

Since the presence of Netrin-1 has been detected in samples of cerebrospinal fluid from patients with brain tumors [104], it is possible to suggest this protein as a useful biomarker to determine the presence of GBM. Moreover, the tumor state could be determined because it has been described that overexpression of Netrin-1 is related to type III and IV gliomas. This would allow the diagnosis of patients with GBM to be much more accurate. Moreover, it would facilitate monitoring those patients who have already received treatment and to know if it has been effective or if there is a recurrence of the disease. Recently, a clinical trial has started the first phase I trial in humans aimed at evaluating the safety, pharmacokinetics, and clinical activity of a humanized monoclonal antibody targeting Netrin-1 (NP137) in patients with advanced/metastatic solid tumors. If the results of this clinical trial were positive, the use of this antibody could be a new option treatment possibility in GBM patients (ClinicalTrials.gov identifier (NCT number): NCT02977195).

## 9. Conclusions

The secretion of angiogenic molecules from tumor cells into the tumor microenvironment facilitates the development of neovascularization in GBM. This review describes the participation of classical angiogenic molecules, as well as Netrin-1, a non-canonical angiogenic molecule, in these processes. Although the role that Netrin-1 plays in GBM has been described, it is necessary to study the way it is secreted by tumor cells and the possible signaling pathways through which it could promote neovascularization in GBM. However, in light of the data presented in this review, it is possible to suggest Netrin-1 as a possible biomarker for GBM. Due to the relationship between its expression and tumor progression, it is possible to speculate that by decreasing its expression, a decrease tumor growth or tumorigenicity of GBM cells could be achieved. The latter could contribute to improve the efficiency of current treatments and, therefore, the survival of patients.

## Figures and Tables

**Figure 2 ijms-22-08248-f002:**
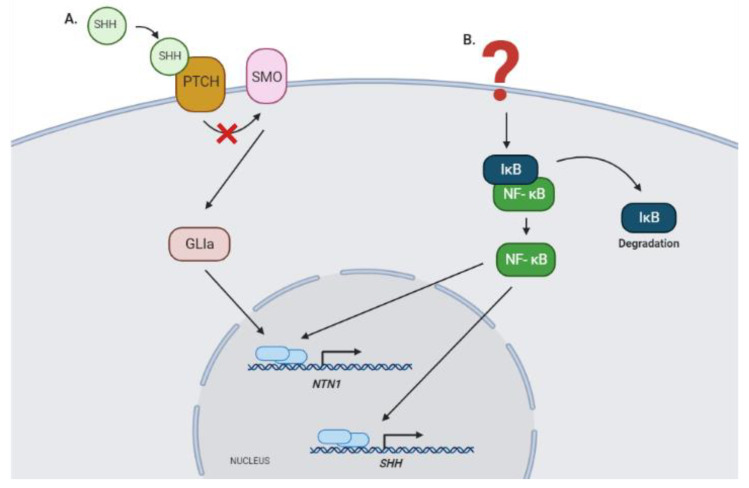
Signaling pathways that could regulate the expression of *NTN1* in GBM. (**A**) *NTN1* expression could be regulated by the canonical SHH signaling pathway, as described recently on medullobastoma, specifically through the transcriptional function of activator isoforms of the GLI zinc finger proteins (Falcon et al., unpublished observations) [72]. (**B**) Alternatively, the NF-κB pathway could modulate NTN1 expression, directly through binding to the promoter region of this gene, or indirectly through the regulation of *SHH* expression [74,75]. GLIa, GLI 1, and GLI2 activator forms; SHH, Sonic Hedgehog protein; PTCH, Patched protein receptor; SMO, Smoothened protein; IκB, inhibitor nuclear factor kappa B; NF-κB, nuclear factor kappa B; NTN1, Netrin-1 gene.

**Figure 3 ijms-22-08248-f003:**
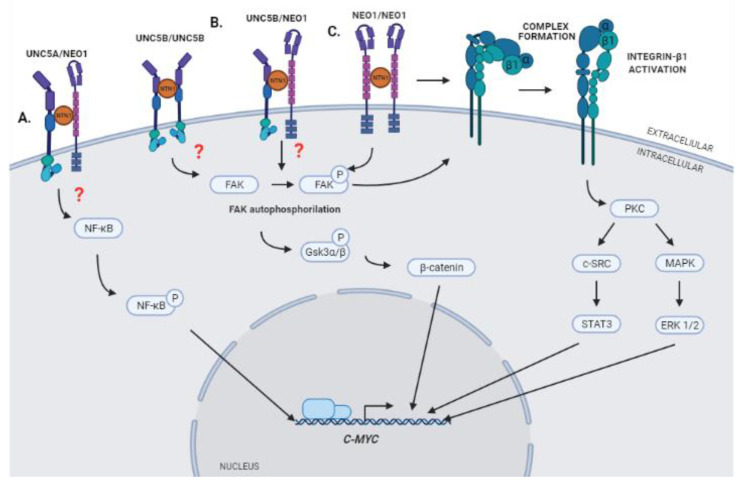
Schematic cartoon summarizing hypothetical models of Netrin-1 signaling that could promote expression of *C-MYC*. (**A**) Interaction of Netrin-1 with its receptor UNC5A may trigger the phosphorylation of NF-κB and the activation of the expression of *C-MYC* [10]. (**B**) Interaction of Netrin-1 with its receptors NEO-1 and UNC5B could promote the phosphorylation of Gsk3α/β and the stabilization of β-catenin, which in turn could trigger the expression of *C-MYC* [84]. (**C**) Interaction of Netrin-1 with its receptor NEO-1 could promote the activation of an Integrin heterodimer, formed by α and β1 subunits, and trigger the expression of *C-MYC* through regulation of the MAPK/SRC axis. STAT3 and ERK 1/2 may promote the induction of *C-MYC* expression by acting as transcription factors in the promoting region of the gene [80].

**Figure 4 ijms-22-08248-f004:**
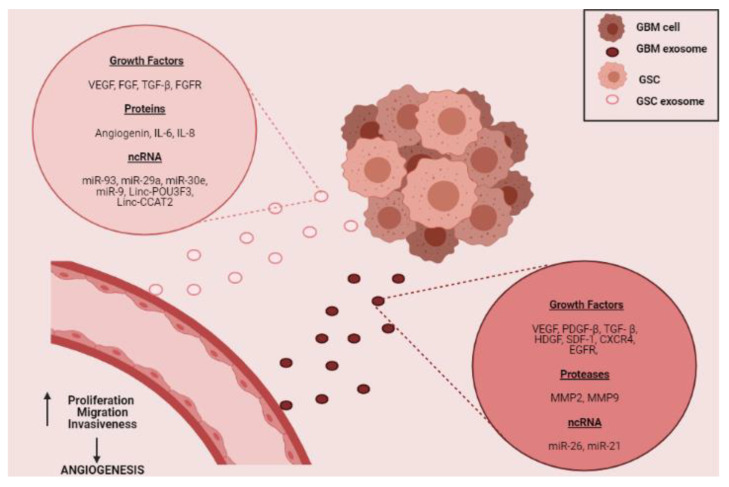
Schematic of possible participation of GBM cell-derived exosomes in promoting angiogenesis in the tumor niche. The figure depicts the possible content of exosomes derived from GSC and/or those derived from GBM tumor cells (growth factors, proteins, proteases, and ncRNAs) [5,6,8,9,89,90,91]. Besides, the uptake of these microvesicles by EC in the tumor vasculature, may promote the increase in the formation of new blood vessels through the promotion of cell proliferation, migration, and invasiveness [6,8,9,91,92,93].

**Table 2 ijms-22-08248-t002:** Netrin-1 and its receptors expression in different cell types of the GBM microenvironment.

Cell Type	Netrin-1	DCC	NEO1	UNC5
Astrocytes	✔ [48]	✔ [50]	✔ [53]	-
Oligodendrocytes	✔ [51]	✔ [51]	-	-
Microglia	✔ [48]	-	-	-
Macrophages	✔ [55]	✔ [53]	✔ [53]	✔ [55]
Lymphocytes	✔ [46]	✔ [46]	✔ [46]	✔ [46]
Brain Endothelial Cells	✔ [49]	✔ [49]	✔ [49]	✔ [54]
Pericytes	✔ [51]	✔ [51]	✔ [51]	✔ [51]

## Data Availability

Not applicable.

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
