# Peer review of "Netrin-1 in Glioblastoma Neovascularization: The New Partner in Crime?"

_ijms, 2021, doi:10.3390/ijms22158248_

Round 1

Reviewer 1 Report

see attached file

Author Response

This clear and overall well-written review should be of interest to reader’s specialist of neo-angiogenesis in cancers.  I have only minor comments aiming at completing or clarifying some of the points addressed by the authors. Comments:

  1. Line 90: The authors wrote “/.../recruitment of bone marrow-derived cells; also known as endothelial precursors/.../”. Since not all bone marrow-derived cells are EC precursors, I believe the authors meant to write “recruitment of bone marrow-derived endothelial precursor cells. «

The phrase has been corrected as suggested by the reviewer (Line 85)

  1. Line 103: Presenting NG2 as a specific mesenchymal lineage marker in the context of brain tumors appears inappropriate In developing and adult central nervous systems NG2 is a marker of oligodendrocyte precursor cells, which are of neural and not mesenchymal origin.

We thank the reviewer for this comment. Following the observation made by Cheng et al (reference 7) we have substituted mesenchymal lineage markers for pericyte markers (Line 97).

  1. Lines 130-131: Is increased expression of VEGFA positively related to poor prognosis when the analysis is restricted to IDH wild-type GBM patients? Inclusion of even a minority of patients bearing a mutant form of IDH can strongly influence the results of such an analysis, with respect to the marked difference in survival expectancy between patients bearing wild-type or mutant IDH1/2 genes.

We agree with the reviewer that mixing wild-type and mutant IDH tumors could be misleading, given the better clinical behavior of the latter. In our recent manuscript (reference 21 in the manuscript), we performed a thorough analysis of the expression of several angiogenic molecules (including VEGF) in relation with some of the most common glioma mutations. Indeed, we found that VEGF expression inversely correlates with the presence of IDHmutations. However, we also found that in IDHwt GBMs, VEGF shows a higher expression in EGFR mutant tumors, which show a worse clinical behavior. Moreover, we determined a vascular signature, which included the VEGFA gene that strongly correlated with a reduced overall survival of glioma patients, both in lower-grade gliomas (enriched in IDHmut tumors) and in GBM. We have included that information in the new version of the manuscript (Lines 124-128).

  1. Line 141: Presenting bFGF as the sole factor promoting maintenance of stem-like state of GBM cells is misleading. GBM cell stem-like state has been shown to be promoted also by EGF PDGF or TGFß for example.

Following the reviewer´s suggestion, we have rephrased that sentence to include those other factors that control GBM stemness (Lines 137-138).

  1. Table 1: PDGF should also be described as one of the factor promoting stem-like properties of GBM cells (for ex. PMID: 25832656). For HDGF the authors wrote that “HDGF Promotes EC migration and angiogenesis in vitro [38].” Since angiogenesis can only be demonstrated in vivo, I verified the article. As expected, angiogenesis was demonstrated using an in vivo assay. Please either remove in vitro or add in vivo.

We thank the reviewer for these comments. We have revised Table 1 to include a new reference for PDGF (reference 31) and we have substituted in vitro for in vivo in relation to HDGF, as indicated.

  1. Lines 177-191: In chapter 4, the authors provide an overview of the known role of Netrin-1 and its receptors in axon guidance prior to discuss available data on Netrin-1 and cancers. With respect to the review’s focus on neovascularization this part should be completed with a summary of the data regarding the participation of this signaling pathway to angiogenic processes in the normal and injured brain (e.g. Fan Y, et al. Overexpression of netrin-1 induces neovascularization in the adult mouse brain. J Cereb Blood Flow Metab. 2008 PMID: 18461079 .Cayre M, et al. Netrin 1 contributes to vascular remodeling in the subventricular zone and promotes progenitor emigration after demyelination. Development. 2013 PMID: 23824572).

We agree with the reviewer that we have neglected the role of Netrin-1 in the process of brain vascularization in normal and pathological conditions. We have included a comment with the two suggested references in the new version of the manuscript (Lines 172-174).

  1. Lines 204-212: The authors describe Netrin-1 expression in GBM cell lines, which have been derived from patients tumors tens of years ago and since then cultured in serum. When transplanted into the brain, these cell lines initiate encapsulated tumors that differ from the usual aspect of patients’ GBM. Recent work performed by the lab where U87 were initially established has moreover demonstrated that the cell line commonly used worldwide did not correspond to the original one (Allen M, Bjerke M, Edlund H, Nelander S, Westermark B. Origin of the U87MG glioma cell line: Good news and bad news. Sci Transl Med. 2016. PMID: 27582061.). For all these reasons, some words of caution should be added when presenting results obtained from this type of GBM cell lines. I would also stretch out that showing that Netrin-1 protein expression is detected by immunohistochemistry in the patients’ tumors would greatly strengthen the proposal that Netrin-1 plays a role in GBM pathology. The authors should look for such evidences and mention them. They might start with the data gathered by the Protein Atlas (www.proteinatlas.org), which provide some pictures of gliomas tissues exhibiting variable Netrin-1 immunostainings.

We thank the reviewer for this sharp comment. We agree that these established glioma cell lines are not always the best model for model for the disease. Following his/her suggestion, we have included a new figure (Figure 1, lines 216-225)) showing representative Netrin-1 stainings in high-grade gliomas (obtained from the Human Protein Atlas) (Figure 1A) (Lines 182-183), as well as RNA-seq data that suggest a stronger expression of this ligand in gliomas in comparison with other cancers (Figure 1B) (Lines 230-231). Moreover, we show that there is a positive correlation between Netrin-1 gene transcription and the expression of several angiogenic-related molecules, including VEGFA (Figure 1C) (Lines 236-240). We strongly believe that these in silico results reinforce the hypothesis of a role for Netrin-1 in gliomas, particularly regulating the vascularization processes.

Reviewer 2 Report

The manuscript entitled “Netrin-1 in glioblastoma neovascularization: The new partner in crime?” by Vasquez et al., is a comprehensive, interesting and well written work about the role of Netrin-1 as a new non-canonical angiogenic factor and putative biomarker . However it needs some ameliorations:

Major points:

  • Pag.1 use of the term “Mixed gliomas”: according to the WHO classification of tumours of the central nervous system, by using both genotype and phenotype as diagnostic criteria, most (if not all) tumours of the group oligoastrocytoma and anaplastic oligoastrocytoma fall into the category of oligodendroglioma or astrocytoma. Only very rare reports describe tumours as “true” oligoastrocytomas, consisting of histologically and genetically discrete astrocytic and oligodendroglial mixed populations of tumour cells. Consequently, accordingly to the CNS WHO 2016 classification criteria, the oligoastrocytoma diagnosis needs to be used only when diagnostic molecular testing is not available, or in the rare real cases where a diagnosis based on histological and molecular genetic features has identified a mixed population of oligodendroglioma and astrocytoma cells in the tumours. In conclusion the definition term “mixed gliomas” should be abandoned or used vary carefully.
  • Legend of Fig.1: in the figure it is reported the “GLIa” term, while in the figure ‘s legend, section (A), GLI1 and GLI2 are described. Is GLIa an acronym of GLI1? Or an acronym of GLI2? Explain/Clarify it better, please.

Minor points:

  • Pag.3 line 134: the hyphen is missing in the word “VEGFR-2”.
  • Table 1: please check the alignment between “protein” and “function”, as it stands now is quite confusing.
  • End of fig.1’s legend: the full stop has been repeated twice.
  • Fig. 3: it is really difficult to read what is written inside the “GBM exosomes” picture, possibly this is due to the dark colour used. Please make that part of the figure more readable.

Author Response

RESPONSE TO REVIEWER 2

The manuscript entitled “Netrin-1 in glioblastoma neovascularization: The new partner in crime?” by Vasquez et al., is a comprehensive, interesting and well written work about the role of Netrin-1 as a new non-canonical angiogenic factor and putative biomarker . However it needs some ameliorations:

Major points:

  • Pag.1 use of the term “Mixed gliomas”: according to the WHO classification of tumours of the central nervous system, by using both genotype and phenotype as diagnostic criteria, most (if not all) tumours of the group oligoastrocytoma and anaplastic oligoastrocytoma fall into the category of oligodendroglioma or astrocytoma. Only very rare reports describe tumours as “true” oligoastrocytomas, consisting of histologically and genetically discrete astrocytic and oligodendroglial mixed populations of tumour cells. Consequently, accordingly to the CNS WHO 2016 classification criteria, the oligoastrocytoma diagnosis needs to be used only when diagnostic molecular testing is not available, or in the rare real cases where a diagnosis based on histological and molecular genetic features has identified a mixed population of oligodendroglioma and astrocytoma cells in the tumours. In conclusion the definition term “mixed gliomas” should be abandoned or used vary carefully.

We thank the reviewer for this comment. It is true that in the new classification of gliomas the term mixed gliomas or oligoastrocytoma is restricted to cases where no molecular or immunohistochemical analysis could be performed. We have simplified the sentence and left only ther terms astrocytomas, oligodendrogliomas and ependymomas (Line 31).

  • Legend of Fig.1: in the figure it is reported the “GLIa” term, while in the figure ‘s legend, section (A), GLI1 and GLI2 are described. Is GLIa an acronym of GLI1? Or an acronym of GLI2? Explain/Clarify it better, please.

Following the reviewers’ suggestion, we have rephrased the figure legend (now Figure 2) to explain that GLIa is the acronym for the activator isoforms of GLI1 and GLI2 (Lines 283-284).

Minor points:

  • Pag.3 line 134: the hyphen is missing in the word “VEGFR-2”.

This mistake has been corrected (Line 131)

  • Table 1: please check the alignment between “protein” and “function”, as it stands now is quite confusing.

We have changed the colors in Table 2 to make it easier to correlate the molecule and the function.

  • End of fig.1’s legend: the full stop has been repeated twice.

This mistake has been corrected (Figure 2´s legend)

  • Fig. 3: it is really difficult to read what is written inside the “GBM exosomes” picture, possibly this is due to the dark colour used. Please make that part of the figure more readable.

We have changed the background color in in the exosomes to facilitate reading the content in the novel Figure 4.